# Incompatibility between two major innovations shaped the diversification of fish feeding mechanisms

**Nick Peoples**[1]*, **Michalis Mihalitsis**[1,2], **Peter C. Wainwright**[1]

**1** Department of Evolution and Ecology, University of California, Davis, California, United States of America, **2** Marine Laboratory, University of Guam, Mangilao, Guam, United States of America

\* npeoples@ucdavis.edu

## Abstract

Innovations often shape the trajectory of macroevolution, yet their effects are usually considered independently, thus ignoring the functional and evolutionary interactions between them. Two innovations that have underpinned the ecological and evolutionary success of ray-finned fishes (Actinopterygii) are large teeth and highly protrusible jaws, which independently expanded the diversity of prey capture strategies. Here, we explore the functional relationship between these innovations across actinopterygians using high-speed videography and phylogenetic comparative methods. We find that these two innovations are functionally and evolutionarily incompatible because there is an overarching tradeoff between jaw protrusion and tooth size. Having large teeth decreases the kinematic diversity of prey capture by restricting species to overtake prey predominantly by swimming, while highly protrusible jaws are only found in species with small teeth. The space within tooth-bearing bones may impose this constraint, by limiting the maximum tooth size of species with gracile jaws adapted for high mobility and jaw protrusion. Nevertheless, some species break this constraint on tooth size through novel adaptations that accommodate exceptionally large teeth, unlocking new feeding modes which may have expanded the nature of aquatic feeding and influenced the ecosystems themselves. Although both high jaw protrusion and large teeth separately expanded prey capture strategies in fishes, they are generally not found in combination and are evolutionarily incompatible.

## Introduction

Innovations, or evolutionary novelties that confer substantial gains in organismal performance, are thought to be a major generator of biodiversity. By allowing species to interact with the environment in new ways, these traits often have profound effects on patterns of ecological, phenotypic, and lineage diversification [1–3]. New adaptive zones and axes of diversification may be unlocked by increased performance, a

**Data availability statement:** All relevant data and code are available within the paper and its Supporting Information files.

**Funding:** This work was supported by the University of California, Davis College of Biological Sciences (funding to P.C.W.). The funders had no role in study design, data collection and analysis, decision to publish, or preparation of the manuscript.

**Competing interests:** The authors have declared that no competing interests exist.

**Abbreviations :** BM, Brownian motion; MCMC, Markov chain Monte Carlo; OU, Ornstein–Uhlenbeck; PGLS, phylogenetic generalized least-squares.

famous example being the modified pharyngeal jaw of some fishes allowing them to process tougher prey items [4]. However, the evolution of functional traits is subject to both biomechanical limits and performance trade-offs, which themselves can have impacts on macroevolutionary patterns [5–9]. Also, the effect of nested innovations may magnify positive effects or lead to conflicts. For example, teeth are a major vertebrate innovation that had extensive consequences for trophic diversity, but the evolution of powered flight in Aves placed a premium on the body being lightweight; this conflict may underlie the loss of teeth in birds [10,11]. Despite the potential for interactions between separate innovations being complex and having a significant effect on macroevolutionary patterns, these interactions are rarely studied.

Ray-finned fishes (Actinopterygii) are the largest vertebrate group in aquatic environments, representing about half of all vertebrates. With over 30,000 species, their success is often attributed to a suite of innovations in the feeding apparatus that have evolved over 400 million years to overcome the challenges of feeding in a fluid [12–15]. Since the Devonian, the evolution of large teeth has provided a multitude of functional benefits that allow fishes to feed across the span of the food web by fracturing tough prey, impaling and restraining evasive prey, or removing prey from a substrate [16–20]. For example, within predatory fishes, species with raptorial teeth often capture prey by grabbing, inflicting damage through laceration and puncture [21,22]. Large teeth with multiple cusps are also characteristic of many early herbivores [23], suggesting that large teeth can reflect feeding specializations across multiple trophic levels.

However, irrespective of prey type, capturing food resources in an aquatic environment requires consumers to be near their prey. Fishes close this gap using a combination of high-speed bursts of forward swimming (body ram), rapid forward projection of their jaws (jaw ram), and hydrodynamic forces (suction) [24,25]. This diversity of prey capture modes reflects further innovations in the feeding apparatus that have improved the efficiency of aquatic feeding through time [13–15,26]. The emergence of suction feeding was a major advance in fish feeding [15,27]. Suction feeding draws-in elusive prey items through a swift flow of water into the mouth, which is generated by rapid expansion of the mouth and oral cavity [28,29]. While many fishes can produce small amounts of suction, the evolution of a protrusible upper jaw was a second major innovation in the feeding system that has since provided multiple benefits to fishes [13–15]. By mechanically increasing hydrodynamic suction forces and swiftly closing the distance to the prey, rapid protrusion of the jaws enhances performance in suction feeding predators [30,31]. Additionally, jaw protrusion enabled fishes to feed on cryptic prey associated with the benthos or elusive prey with fast escape mechanisms [26]. Upper jaw protrusion has evolved independently multiple times with different mechanisms but shows convergent effects on function [14,32,33]. Protrusible jaws and large teeth are thus two innovations that both improved the efficiency of feeding modes and fundamentally expanded the diversity of prey capture strategies in fishes, which may have contributed to their dominance in the aquatic realm [12–14,20,26,27]. However, the functional and evolutionary relationships between these two innovations remain unclear.

In this study, we examined the relationship between tooth size and the kinematic diversity of feeding strikes across a diverse array of ray-finned fishes (Actinopterygii) to understand the functional and evolutionary dynamics between these two major advances in aquatic feeding. We captured high-speed feeding videos of 161 actinopterygian species spanning 23 orders and 66 families to measure the contributions of body ram, jaw protrusion, and suction to prey capture and assessed these relationships with tooth size in a phylogenetic framework. We fit continuous trait evolution models to describe macroevolutionary patterns of tooth size evolution. If fishes can capitalize on both innovations – large teeth and jaw protrusion – we expect to find species with large teeth using high amounts of jaw ram during prey capture. Our results provide key insight into the functional and evolutionary relationships between two major innovations that drove success within the largest group of aquatic feeding vertebrates.

## Results and discussion

### Tooth size and the kinematic diversity of feeding

We found extensive variation in relative tooth size in the oral jaws of 161 species of ray-finned fishes, spanning 23 orders and 66 families (S1 Table; Fig 1A). Tooth size varied 160-fold across the phylogeny and ranged from being absent (e.g., *Neolissochilus stracheyi*, blue mahseer) or significantly reduced (e.g., *Lates niloticus*, Nile perch) to long piercing fangs (e.g., *Hydrolycus armatus*, vampire tetra) and forceps-like incisors (e.g., *Zanclus cornutus*, Moorish idol). The largest and smallest teeth are found in old lineages (Fig 1A) and there is high phylogenetic signal of tooth size across the phylogeny (Pagel's $\lambda = 0.94$; $P < 0.0001$), suggesting that long evolutionary periods are required for significant change in tooth size.

We quantified the diversity of prey capture strategies using high-speed video to estimate the relative contribution of body ram, jaw ram, and suction to closing the gap between the fish and its prey. The kinematic diversity of feeding strikes ranged from the exclusive use of body ram (e.g., *Boulengerella maculata;* body ram relative contribution [$P_{br}$] = 0.998, jaw ram relative contribution [$P_{jr}$] = 0.001, suction relative contribution [$P_{s}$] = 0.001) to a primary contribution of jaw ram (e.g., *Aulostomus maculatus*; $P_{br} = 0.021$, $P_{jr} = 0.938$, $P_{s} = 0.041$) or suction (e.g., *Odontanthias borbonius*; $P_{br} = 0.121$, $P_{jr} = 0.428$, $P_{s} = 0.451$) (Fig 1A). The majority (76%) of species close the distance to the prey predominantly through forward swimming (body ram). We used phylogenetic generalized least-squares (PGLS) regression to determine relationships between tooth size and the proportions of body ram, jaw ram, and suction used during prey capture while accounting for phylogenetic relatedness. Tooth size has a significant relationship with the relative contributions of body and jaw ram but not suction (Fig 2A). There is a strong positive relationship between tooth size and the proportion of body ram used during feeding (PGLS; $t = 3.707$, $P < 0.001$). As tooth size increases, species are restricted to using greater amounts of body ram to capture prey, and species with the largest teeth are almost exclusively body ram feeders (Fig 2A). Conversely, we find a significant negative relationship between tooth size and the contribution of jaw ram (PGLS; $t = -4.999$, $P < 0.001$). A reduction in tooth size appears to be necessary for the highest use of jaw ram during prey capture, but species with reduced tooth size can still use high amounts of body ram (Fig 2A). Extreme cases of these reductions in tooth size include the complete loss of teeth within toothed families (e.g., *Emmelichthyops atlanticus* [Haemulidae]; $P_{jr} = 0.678$) [34] or the evolution of edentulism (i.e., absence of teeth) in coordination with a novel form of jaw protrusion as seen in carps and minnows (Cyprinidae) [14,35]. To examine the relationship between jaw protrusion and tooth size at a broader phylogenetic scale, we compared tooth size between two large clades in our study that differ in their capacity for jaw protrusion. Though Characiformes (tetras, piranhas, headstanders), a clade that largely lacks premaxillary jaw protrusion [36,37], have 1.6× larger teeth than cichlids, which all have protrusible jaws, this difference is not significant when accounting for unequal group variances (Welch's Analysis of Variance [ANOVA]; $F = 2.32$, $P = 0.146$) (S2B Fig). We find no relationship between tooth size and the relative contribution of suction (PGLS; $t = 0.287$, $P = 0.775$) (Fig 2A). Though produced by most ray-finned fishes, suction often has a minor contribution to closing the predator-prey distance, relative to jaw ram and body ram. In our data, only three species (2% of all species analyzed) had a suction contribution of >50%, and 37% of species had a negligible (<10%) contribution of suction, confirming body ram-jaw ram as the primary axis of feeding diversity in

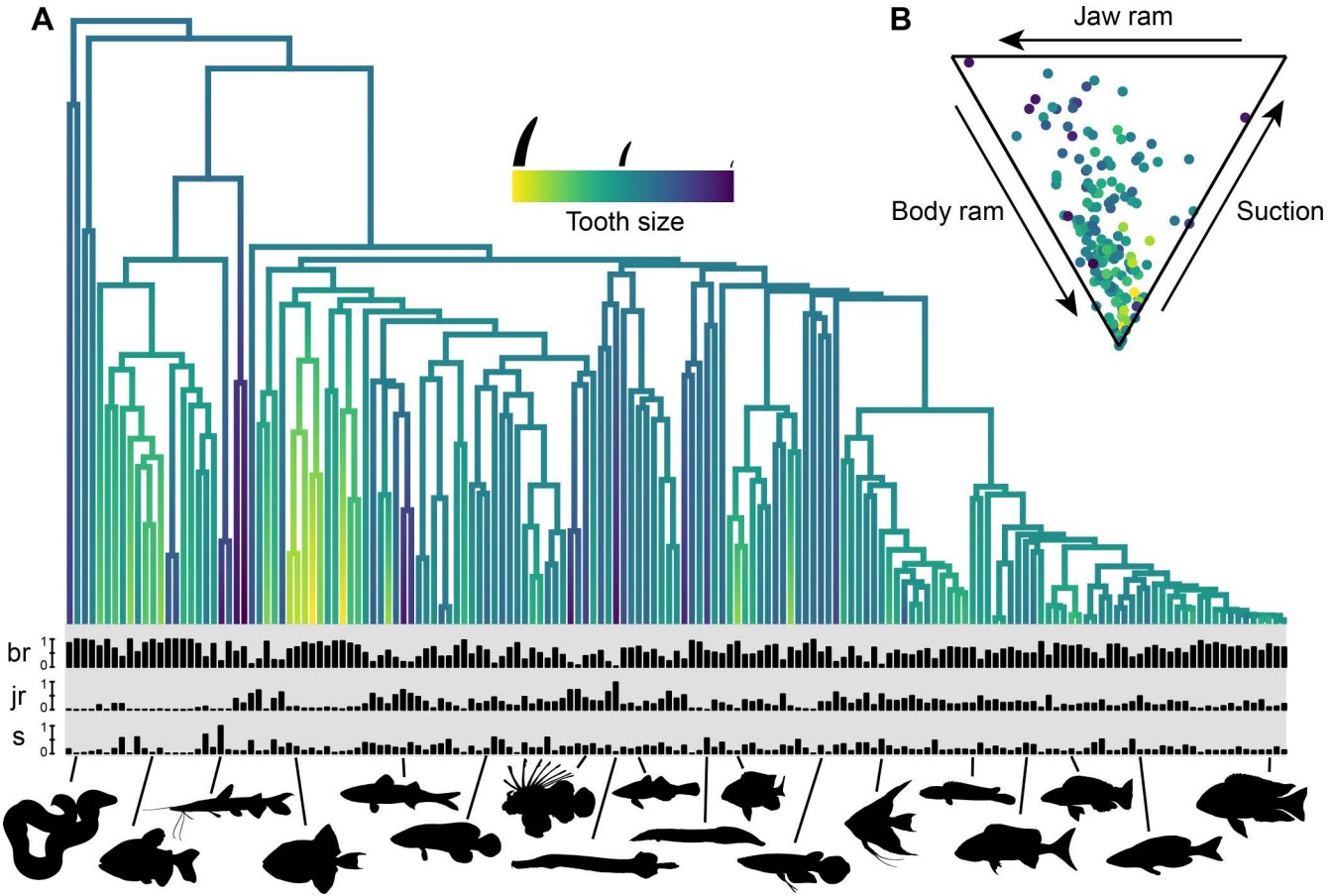

**Fig 1. The diversity of tooth sizes and feeding kinematics across ray-finned fishes. (A)** The phylogenetic distribution of tooth size across 161 species of ray-finned fishes. Barplots at the tips correspond to the proportion of body ram (br), jaw ram (jr), and suction (s) for each species. Species (from L to R): *Gymnothorax griseus, Piaractus brachypomus, Sorubim lima, Pseudobalistes fuscus, Emmelichthyops atlanticus, Rypticus maculatus, Pterois volitans, Aulostomus maculatus, Butis butis, Mastacembelus armatus, Microspathodon chrysurus, Aplocheilus lineatus, Pterophyllum scalare, Teleogramma brichardi, Boulengerochromis microlepis, Julidochromis dickfeldi, Cyprichromis leptosoma, Chilotilapia rhoadesii.* **(B)** Ternary plot depicting the combined proportions of body ram, jaw ram, and suction; each point represents a species, and the color of the points corresponds to tooth size. The data underlying this figure can be found in S1 Data and S2 Data.

fishes [24] (Fig 1A, 1B). We tested the sensitivity of these results to the large sample of cichlids ($n = 59$ species) in our dataset and found concordant patterns when all cichlids were removed before analysis (PGLS no cichlids; $P_{body\ ram} < 0.001$, $P_{jaw\ ram} < 0.001$, $P_{suction} = 0.053$) (S1 Fig).

While PGLS can account for phylogenetic relatedness, proportional data is often non-linear [38]. To supplement our PGLS analyses, we fit Dirichlet regression models that can effectively model non-linear relationships between proportional data with more than two categories [38–40]. Model comparison using maximum likelihood confirmed that tooth size can predict the relative contribution of body ram and jaw ram in a strike, but not of suction ($\Delta AICc = 2.12$) (S2 Table). As tooth size increases, the proportion of body ram used to capture prey increases while the contribution of both suction and jaw ram decreases (Fig 3A), which follows the patterns of the PGLS results. Overall, our results show an overarching trade-off axis between different feeding modes being used by ray-finned fishes with different tooth sizes.

The functional diversity of prey capture for lineages with large teeth may be limited if large teeth constrain species to body ram feeding. To quantify the diversity of prey capture for multiple ranges of tooth size, we measured the variance

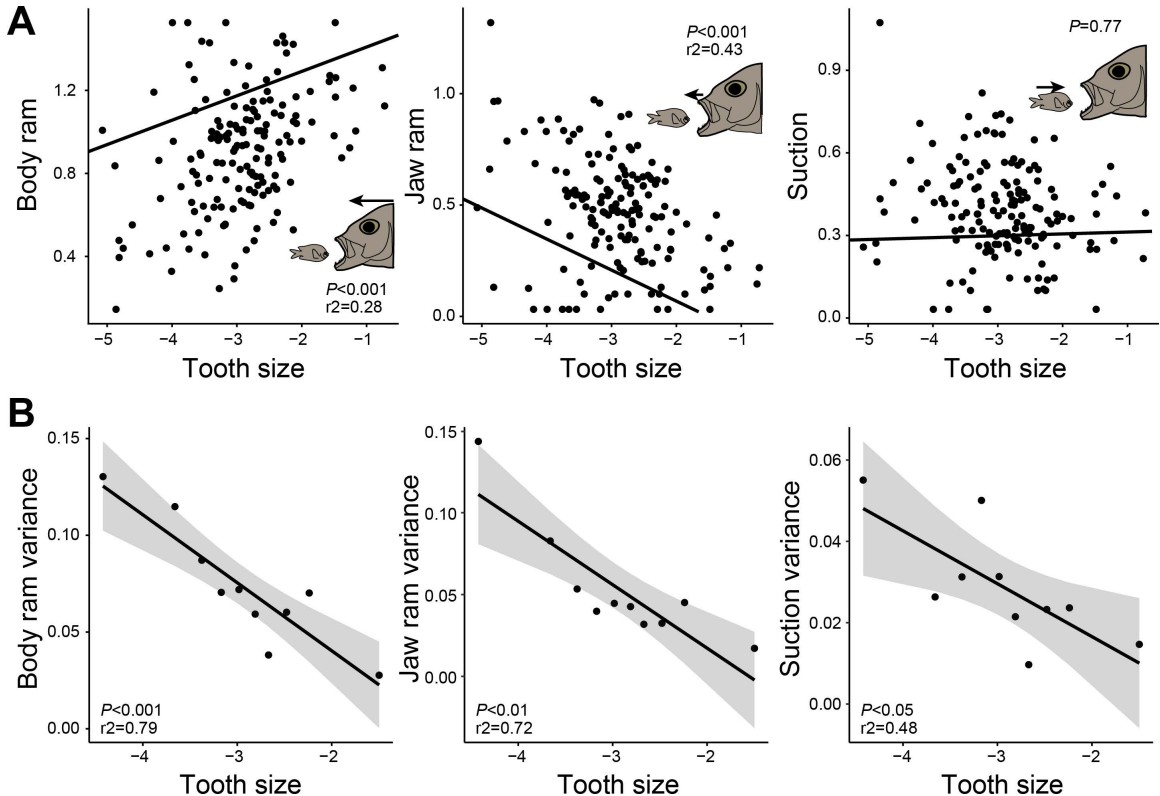

**Fig 2. The relationship between prey capture kinematics and tooth size in ray-finned fishes.** (A) Phylogenetic generalized least-squares (PGLS) regressions between tooth size, quantified as a log-shape ratio, and the arcsine-transformed proportions of body ram, jaw ram, and suction. *P*-values and $r^2_{pred}$ are reported for each regression. (B) Linear regressions between kinematic variance and tooth size, for body ram, jaw ram, and suction. Ninety-five percent confidence intervals are depicted in grey. Each point corresponds to the variance quantified from 10 equal-frequency bins, based on tooth size. The data underlying this figure can be found in S1 Data and S2 Data.

in body ram, jaw ram, and suction contributions for 10 equal-frequency tooth-size bins. Linear regression models reveal the variances of all three components have significantly negative relationships with tooth size (body ram: $df = 8$, $F = 34.68$, $P < 0.001$, $r^2 = 0.79$; jaw ram: $df = 8$, $F = 24.37$, $P < 0.01$, $r^2 = 0.72$; suction: $df = 8$, $F = 9.283$, $P < 0.05$, $r^2 = 0.48$) (Fig 2B). Variance in the contribution of these components to prey capture is 3.7–8.4x lower between species with the largest and smallest teeth. These consistent linear decreases in variance across species show that the diversity of prey capture kinematics decreases as teeth become larger. We recover this strong relationship between feeding kinematics and tooth size in spite of extensive ecological diversity among large-toothed species. These include grabbing piscivores (*Hydrolycus armatus*), zooplanktivores (*Ecsenius midas*), mobile invertivores (*Rhinecanthus rectangulus*), and herbivores (*Zebrasoma flavescens*). Although having large teeth in the jaws may constrain species to body ram feeding, small teeth permit species to capture prey using a greater diversity of kinematic strategies by utilizing varied amounts of jaw ram and suction.

## Models of tooth size evolution

As species that use extensive jaw protrusion to capture prey are constrained to have small teeth, we hypothesized that adaptive evolution 'pushes' jaw ram feeders to have smaller teeth than body ram feeders. We tested this by fitting continuous trait evolution models of tooth size evolution, after classifying species as either jaw ram or body ram feeders. Model comparison reveals that tooth size evolves under an Ornstein–Uhlenbeck (OU) process with distinct optimal tooth sizes

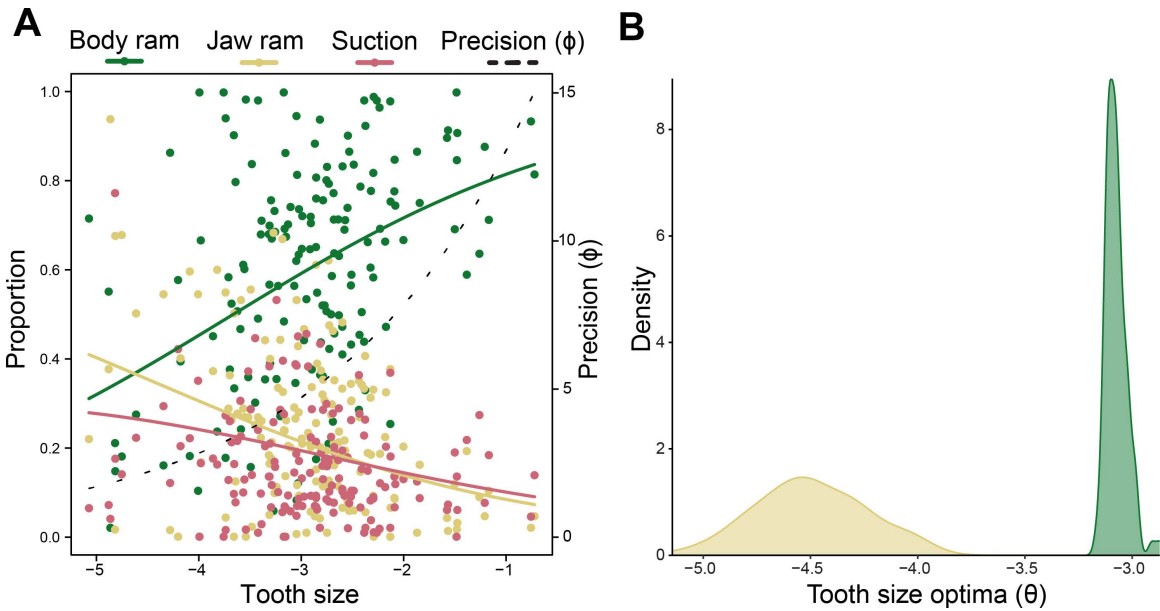

**Fig 3. Large teeth and highly protrusible jaws are evolutionarily incompatible. (A)** Dirichlet regression between the proportions of body ram (green), jaw ram (yellow), and suction (red) used in a strike, and tooth size, quantified as a log-shape ratio. Precision ($\phi$) is plotted on the second $y$-axis (dotted line). **(B)** Distribution of optimal values for log-transformed tooth size ($\theta$) between body ram (green) and jaw ram (yellow) feeders under a multiple optima Ornstein–Uhlenbeck model with single $\sigma^2$ and $\alpha$ parameters (OUM), based off 100 model fits. The data underlying this figure can be found in S1 Data and S3 Data.

for body and jaw ram feeders, but with the same evolutionary rate ($\sigma^2$) and strength of selection ($\alpha$) ($\Delta$AIC = 0.74) (S3 Table; Fig 3B). The small size of our dataset reduces the power to distinguish between models that allow $\sigma^2$ and $\alpha$ to vary, however there was strong support for all models that allowed distinct evolutionary optima for jaw and body ram feeders when compared to Brownian models and a single-optimum OU model (OU1; S3 Table). We find a relationship where the optimum tooth size for jaw ram feeders (body size-independent tooth size [shape ratio] = 0.011) is 4.1× smaller than that of body ram feeders (shape ratio = 0.046) (Fig 3B). These optima are similar to the observed tooth sizes of *Monocirrhus polyacanthus* (Amazon leaffish, 59.6% jaw ram) and *Gymnothorax griseus* (geometric moray eel, 99.8% body ram). We also estimated the evolutionary correlation ($r$) between jaw ram feeding and tooth size under a threshold model, in which the state of a discrete character (i.e., jaw ram feeding) changes depending on the value of a secondary character (i.e., tooth size) [41]. We find a negative evolutionary correlation between increasing tooth size and feeding through jaw ram ($r = -0.436$), indicating that an antagonistic interaction defines the evolution of these two traits (S2A Fig). Overall, our results on the evolutionary dynamics between tooth size and feeding mode reveal that tooth size affects the use of jaw protrusion during feeding. Large teeth constrain prey capture kinematics to primarily body ram feeding while the combination of small teeth and jaw protrusion represents a separate adaptive peak.

## Large teeth and highly protrusible jaws are incompatible

Our results reveal that jaw protrusion and large teeth, two notable innovations in the fish feeding system, are functionally and evolutionarily incompatible with each other. Independently, these innovations have greatly expanded the trophic opportunities of fishes in their habitats [14,20,26]. However, they cannot simultaneously be maximized, pointing towards a major trade-off axis and implicating a biomechanical relationship in governing the diversity of prey capture techniques in fishes. Evolution towards large teeth constrains the diversity of prey capture kinematics, requiring

species to use increasingly greater proportions of body ram and thus close the gap between themselves and the prey through swimming. Our results point to adaptive evolution towards reduced tooth sizes as a crucial component in the evolution of fishes with highly protrusible jaws. However, without having large teeth, jaw ram feeders are limited in the way they can detach, manipulate, and damage food resources. This restriction may have evolutionary implications for jaw ram feeders and follows a well-known tradeoff in fishes between force and velocity-modified jaws that is strongly related to ecology [7,42–44]. Species with velocity-modified jaws often use high protrusion and consume prey whole in rapid strikes with little oral manipulation or processing [21,24]. In contrast, species with little to no jaw protrusion often target attached, non-evasive prey that require removal from a substrate or elusive prey that requires further restraint or damage by raptorial teeth. Considerable change in dentition and skeletal elements of the jaw system is likely required for jaw ram feeders with small teeth to undergo dietary transitions towards resources that require detachment or post-capture processing [42].

Biomechanical limits on both feeding performance and the evolution of feeding structures is a widespread feature of vertebrate evolution [45–48], suggesting there may be a biomechanical limit on tooth size. Intraosseous (i.e., within the bone) tooth replacement is a derived state in teleosts and the predominant replacement mode among fishes in our study (e.g., Perciformes, Cichliformes, Characiformes, Acanthuriformes) [49] (S1 Table). By requiring replacement teeth to develop in a crypt within the dentigerous bone [49], this process may impose a limit on tooth size. As replacement teeth cease developing, they erupt and replace the functional tooth in a "one for one" manner, joining the existing functional tooth row [49–51]. However, the dentigerous bone must have enough space to accommodate these full-size replacements. This presents a dilemma for many acanthomorph species with premaxillary protrusion. Fishes with high jaw protrusion that capture prey through jaw ram typically have slender, mobile jaws adapted for rapid motions [42,52], and therefore do not have as much space for teeth to develop within the jaw bone. We suggest that the jaws of species with high jaw protrusion may impose a structural limit on tooth size based on the space available for tooth development in the dentigerous bones. In addition to this constraint, tooth size may be minimized in jaw ram feeders to avoid interference with suction forces or reduce drag during high-velocity strikes [53].

## Unique adaptations to accommodate large teeth

Our results demonstrate that increased tooth sizes limit species to predominantly body ram feeding. While large teeth (in our study, the upper 10% of the tooth size distribution) may limit the kinematic diversity of feeding, they have unlocked a wide range of unique ecological niches by providing unique abilities to damage, detach, cut, and collect food resources. For fishes feeding on non-evasive resources, long teeth are a more recent (approximately 40 mya) adaptation that unlocked new benthic resources including coral mucus and detritus [20,54]. Large teeth have given rise to other specialized feeding modes such as herbivory (e.g., Siganidae, Acanthuridae) [55], wood-eating (e.g., Loricariidae) [56], frugivory (e.g., Serrasalmidae) [57,58], and exceptional feeding behaviors such as entrapping prey in a dental cage, seen in 'wolf-trap' anglerfishes (Ceratioidei) in the deep sea [59]. Despite our results pointing towards a biomechanical limit on relative tooth size, species with exceptionally large teeth (S1 Table) are united by unique, independent adaptations to the modes of tooth attachment and replacement as well as modifications to the teeth themselves (Fig 4A–4D). These modifications allow large dental tools to be positioned in a comparatively small set of jaws. For example, the modified tooth replacement process of pufferfishes (Tetraodontidae) and parrotfishes (Labridae) gives rise to a beak, functionally equivalent to a single large tooth. In parrotfishes, superimposed rows of rounded teeth are cemented together in a bone-like layer (Fig 4A) while in pufferfishes, teeth are continuously stacked together to form a compact, highly-mineralized osteodentine mass [60–62] (Fig 4B). These modifications, which circumvent the common "one for one" replacement in fishes, enabled these species to unlock new ecological roles within their environments [63]. Modification to the location of tooth attachment enables the positioning of larger teeth in the jaw. The procumbent implantation of teeth in *Zanclus cornutus* (Moorish idol) and associated horizontal tooth crypts within the jaw illustrate modifications to the orientation of tooth crypts used to accommodate

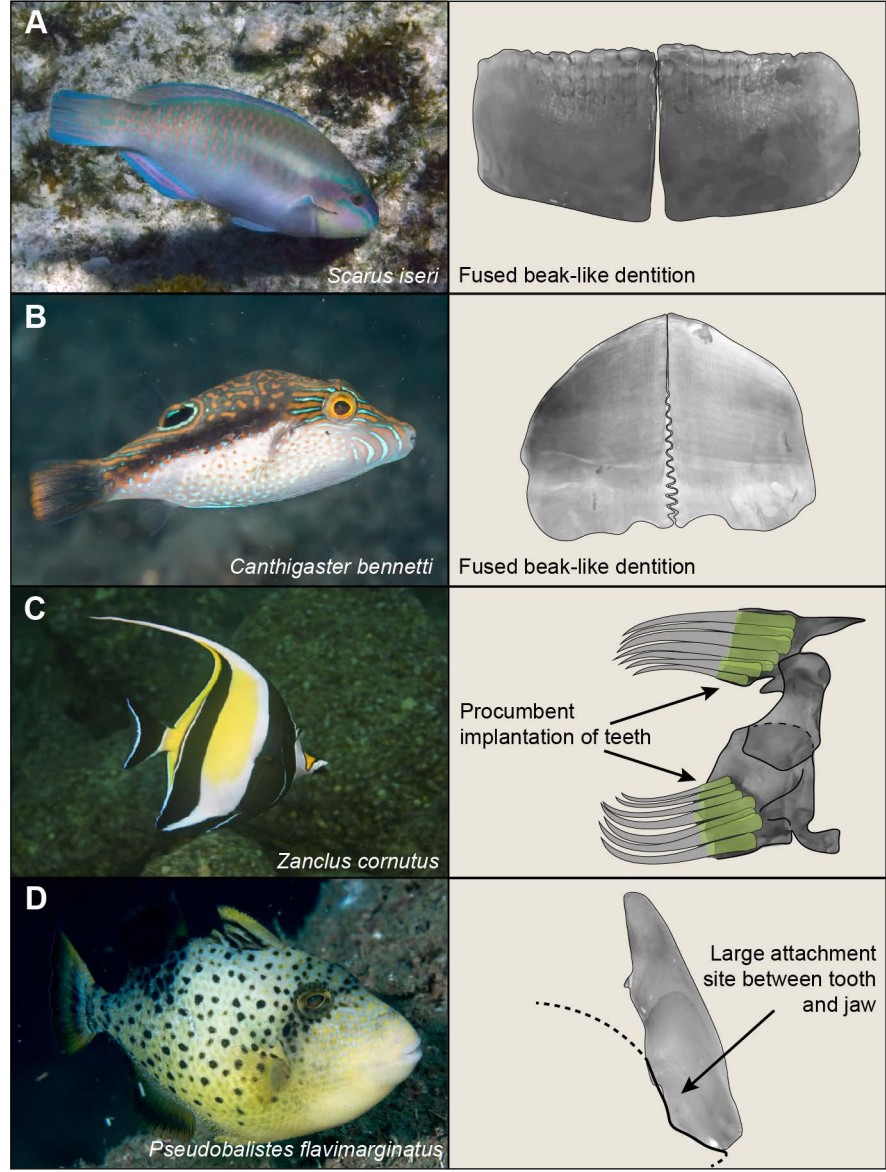

**Fig 4. Extreme morphological adaptations to accommodate large teeth in fish jaws.** Depicted are species from the upper 10% of our dataset on tooth size, which have all evolved unique modifications to accommodate large teeth in their jaws. (A) Upper jaw of *Scarus iseri*, in labial view. Photo credit Louis Imbeau, iNaturalist (CC BY 4.0). (B) Lower jaw of *Canthigaster bennetti*, in labial view, showing the fused beak. Photo credit Rickard Zerpe, Flickr (CC BY 2.0). (C) Lateral view of *Zanclus cornutus* upper jaw showing the procumbent implantation of teeth. The arrow indicates the location of a horizontal tooth crypt. Photo credit Laszlo Ilyes, Flickr (CC BY 2.0). (D) Lateral view of teeth from the lower jaw of *Pseudobalistes flavimarginatus*. The arrows indicate large attachment sites on the lingual aspect of the tooth. Photo credit Rickard Zerpe, Flickr (CC BY 2.0).

larger tooth sizes (Fig 4C). In triggerfishes (Balistidae), tooth attachment occurs between the lingual aspect of the tooth and labial aspect of the jaw with modified connective tissue [64], which provides a larger attachment surface (Fig 4D). An extreme case of modification to the structure of teeth themselves is found in some butterflyfishes (Chaetodontidae), which have highly flexible tooth shafts, a feature similar to the teeth of some loricariid catfishes [65]. Overall, while the reduction of tooth size may be achieved relatively simplistically through conserved developmental pathways [66], coordinated

changes within the tooth developmental program, as well as major changes to the bone morphology of the jaws and skull, may be required for extreme increases in tooth size (Fig 4A–4D).

## Conclusions

Our results demonstrate a fundamental incompatibility between two functional innovations in fish feeding, which separately contributed to the success of fishes in aquatic environments by enabling new prey capture strategies and consequently, altered ecosystem processes [12,20,26]. We use high-speed videos to describe the kinematic diversity of fish feeding, finding strong relationships between tooth size and the major axis of prey capture diversity in fishes [24]. Evolutionary model fitting reveals that adaptive evolution towards small teeth is a feature of jaw ram feeders with negative implications for resource acquisition abilities [42,67]. Enlarged teeth facilitate novel trophic habits but limit kinematic diversity while small teeth permit extensive kinematic diversity yet limit resource processing abilities. The evolution of large teeth may highlight major transitions in the history of fishes where despite restricting kinematic diversity and requiring major modifications, it allowed them to explore novel niches and previously inaccessible resources in competitive habitats, such as coral reefs. While feeding innovations may profoundly expand the trophic opportunities of fishes, underlying trade-offs can impose functional and evolutionary incompatibilities between these traits, preventing simultaneous expression of multiple innovations.

## Materials and methods

### Ethics statement

All experiments were carried out in accordance with the University of California, Davis Institutional Animal Care and Use Committee, protocol numbers 22206 and 23818.

### High-speed video

During feeding events, fish close the distance between themselves and prey through the use of three mechanisms: (1) they draw the prey towards their mouth using suction, (2) they move their body toward the prey by swimming (body ram), or (3) they extend their mouth toward the prey using jaw protrusion or flexion of the body (jaw ram) [24,25]. We quantified the proportions of these three movements (body ram, jaw ram, suction) to closing the total distance between predator and prey during feeding events of 161 species of ray-finned fishes (Actinopterygii) across 66 families (S1 Table). 140 species were filmed for this study and data for 21 species was taken from [24]. Individuals were fed a variety of foods to elicit strikes, including brine shrimp, live black worms, or live feeder fish (*Gambusia affinis*); we acknowledge that incorporating a variety of prey types may introduce some variation in the relative use of ram and suction during the strike. Fish were filmed in 110–210-liter aquaria, depending on their size.

Individuals were recorded feeding using digital high-speed cameras (Fastec Imaging, High Spec 2G or Photron Fast-Cam Mini) at 1,000−2,000 frames per second (fps). Multiple strikes were recorded for each individual, and the single best strike (i.e., limited movement of the prey and lateral positioning of the fish relative to the camera throughout the strike) was used for further analysis. All sequences were chosen by a single observer (N.P.) to minimize bias in the choice of strike for analysis. Following filming, each specimen was euthanized with overexposure to a solution of MS-222 and fixed in buffered formalin. For each sequence, we extracted two frames using the software ImageJ [68]; (1) the onset of the strike, and (2) the time at which the prey's estimated center of mass passed through the mouth of the predator. We determined the onset of the strike to be at the start of depression of the buccal cavity, cranial elevation, or acceleration towards the prey, which varied between species.

### Quantifying kinematic proportions

To quantify the proportions of each kinematic component used in each strike, we rotated and aligned the two frames for each strike in Adobe Illustrator (Adobe), so that the strike occurred along the *x*-axis of the aligned frames. To quantify body

ram, we measured the distance between the center of the eye between frames. Jaw ram was quantified as the length of the protruded jaw in the second frame, measured from the tip of the maxilla, which was often the point of insertion of the premaxillary ascending process, to the anteriormost point of the upper jaw. This measurement includes a negligible constant (compared to the contribution of other kinematic components) for each species, which represents the distance between these points in a closed jaw system. Suction was measured as the horizontal distance travelled toward the predator by the center of mass of the prey item between the two frames. Each measurement was then converted to a proportion of the total movement (body ram + jaw ram + suction).

## Measuring tooth size

We extracted the single largest tooth of each individual under a dissecting scope, mounted each tooth flat on a slide with double sided tape, and photographed it with a microscope camera (Hangzhou ToupTek Photonics Co., Ltd). We measured tooth size as the linear distance between the midpoint of the base of the tooth and the tip of the cusp using the software ToupView. To capture the size of the individual, we measured standard length (SL), upper jaw length (JL; measured from the tip of the premaxilla to the end of the maxilla), and jaw width (JW; between ends of the maxilla) using dial calipers. We size-corrected tooth length using log shape ratios, using the geometric mean of SL, JL, and JW to represent body size [69]. We +0.1 transformed all raw measurements of tooth length before calculating the geometric mean to accommodate edentulate species (tooth length = 0 mm). This method of size correction accounts for the wide range of body shapes and relative jaw sizes in our dataset. Two substitutions were made based on specimen availability: *Rhinecanthus verrucosus* (tooth size measured) for *Rhinecanthus rectangulus* (filmed), and *Pseudobalistes flavimarginatus* (tooth size measured) for *Pseudobalistes fuscus* (filmed).

## Quantification and statistical analysis

We conducted all analyses in R v4.2.2 (R Core Team, 2022). For all phylogenetic analyses, we used the DNA sequence based phylogeny of [70], pruned to include the species in our study ($n = 161$ species). We substituted *Dormitator latifrons* for *Dormitator lebretonsis* (filmed in this study), as *D. lebretonsis* was absent from the molecular phylogeny. First, we applied an arcsine transformation to the proportions of body ram, jaw ram, and suction to improve normality. To examine the relationship between the contribution of each component and tooth size, we used phylogenetic generalized least squares regression (PGLS) implemented with the '*gls*' function in the package '*nlme*' (v.3.1–160) [71]. We accounted for phylogenetic relatedness by setting a correlation structure under Brownian motion ('*corBrownian*' in the package '*ape*' (v.5.8) [72]. For each model, we estimated the variance explained by tooth size by calculating $r^2$ for predication ($r^2_{pred}$) in the package '*rr2*' (v.1.1.1) [73,74]. To test the sensitivity of our results to our sampling bias towards Cichlidae ($n = 59$ species), we repeated these analyses after excluding all cichlids.

We supplemented our PGLS analyses with Dirichlet regression using the '*DirichletReg*' package (v.0.7–1) [40]. Although this approach does not account for phylogenetic relationships, it can model non-linear relationships between tooth size and all three kinematic proportions together, providing an advantage when used together with PGLS. We compared the fits of six models using AICc scores (for a description of the models, see S2 Table). The significance of the PGLS analyses suggests that while tooth size alone may have high phylogenetic signal, its relationship with body and jaw ram is much less dependent on phylogenetic relationships. Therefore, we would not expect the general patterns or significance of the Dirichlet regression results to change if phylogeny was included. To test for a relationship between the diversity of kinematic proportions used during feeding and tooth size, we first discretized species into 10 equal-frequency ($n = 16$) bins (one bin $n = 17$) based on tooth size using the package '*discretize*' function in the '*arules*' package (v.1.7.8) [75]. For each bin, we calculated the mean tooth size. We then fit linear regression models with the '*lm*' function in the '*stats*' package (v.4.2.2), where variance is a function of tooth size. We then calculated $r^2_{pred}$ for each regression to estimate the variance explained.

PLOS Biology

To understand how tooth size evolves across ray-finned fishes, we fit both Brownian motion (BM) and Ornstein–Uhlenbeck (OU) models of trait evolution in 'OUwie' (v.2.10) [76]. We categorized species as either body or jaw ram feeders (a binary trait) based on the highest contribution, ignoring suction, given our initial results. For example, if a species' contributions were br = 0.25, jr = 0.35, and s = 0.4, that species was categorized as a jaw ram feeder. We removed suction for this categorization as we found no relationship between tooth size and suction, and only a small number of species fed predominantly through suction. We generated 100 stochastic character maps under an equal-rates (ER) transition model using 'make.simmap' in 'phytools' (v.2.1.1) [77] to account for uncertainty in the history of body and jaw ram feeding. We then fit seven trait evolution models (S3 Table) across all 100 simmaps and compared models using the mean AIC scores. The small size of our dataset reduces the power to distinguish between models that allow the evolutionary rate ($\sigma^2$) and strength of selection ($\alpha$) to vary, however there was strong support for all models that allowed distinct evolutionary optima for jaw and body ram feeders when compared to Brownian models and a single-optimum OU model (OU1; S3 Table). We fit an additional threshold model of trait evolution in a Bayesian framework using the 'threshBayes' function in 'phytools' to evaluate the strength of evolutionary covariation ($r$) between tooth size and jaw ram feeding [41]. The Markov chain Monte Carlo (MCMC) was run for 500,000 generations, sampling every 500 generations, and we removed the first 10% of samples as burn-in. We estimated phylogenetic signal in our tooth size data using 'phylosig' in 'phytools' (method = "lambda"). We compared tooth size between two large clades in our study, cichlids ($n$ = 59 species) and characiformes ($n$ = 16 species) using Welch's ANOVA in the R package 'rstatix' (v.0.7.2) [78] to account for unequal sample variances (Levene's test; $df$ = 74, $F$ = 10.036, $P$ < 0.01).

## Supporting information

**S1 Fig. The relationship between prey capture kinematics and tooth size in ray-finned fishes, after removing all Cichlidae species.** Phylogenetic generalized least-squares (PGLS) regressions between tooth size, quantified as a log-shape ratio, and the arcsine-transformed relative contributions of body ram, jaw ram, and suction, for 102 species of ray-finned fishes, after removing all 59 species of cichlids. $P$-values and $r^2_{pred}$ are reported for each regression. The data underlying this figure can be found in S1 Data and S2 Data.
(TIF)

**S2 Fig. Evolutionary relationships between tooth size and jaw protrusion. (A)** Posterior distribution of the evolutionary correlation coefficient ($r$) between tooth size and body and jaw ram feeding, estimated under a threshold model. The blue dotted line indicates the mean of the distribution. **(B)** Clade comparisons of tooth size between Characiformes (no premaxillary protrusion) and Cichlidae (premaxillary protrusion). Significance is reported from Welch's Analysis of Variance (ANOVA) test. The data underlying this figure can be found in S1 Data and S4 Data.
(TIF)

**S1 Table. List of taxa included in this study.** A complete list of taxa included in this study. Species with exceptionally large teeth, defined as the upper 10% of the tooth size distribution, are marked with a (*).
(PDF)

**S2 Table. Dirichlet regression model fitting results.** Dirichlet regression model fitting using the 'DirichReg' function in the R package 'DirichletReg'. 'dr' is the compositional data of the proportions of body ram, jaw ram, and suction, and is tooth size, corrected for body size. AIC scores, corrected for small sample size, are reported. The best-fit model is bold.
(PDF)

**S3 Table. Ornstein–Uhlenbeck model fitting results.** Results of continuous trait evolution model fitting with 'OUwie'. AIC scores, corrected for small sample size, are reported as the mean of 100 fitted models. The best-fit model (OUM) is

identified in bold. Parameters refer to the Brownian rate parameter ($\sigma^2$), strength of selection ($\alpha$) and location of the adaptive peak ($\theta$).
(PDF)

**S1 Data. Data generated and analyzed during this study.** Data on tooth size and the proportions of body ram, jaw ram, and suction used during feeding for 161 species of ray-finned fishes.
(CSV)

**S2 Data. Time-calibrated phylogeny of 161 ray-finned fish species.**
(NEXUS)

**S3 Data. Output of 100 fitted Ornstein–Uhlenbeck models.** Optimum tooth sizes for body ram and jaw ram feeding species.
(CSV)

**S4 Data. Output of threshold model of character evolution.**
(CSV)

**S1 Code. R code for all phylogenetic comparative analyses.**
(R)

## Acknowledgments

We wish to thank H. Chan, B. Zhang, and N. Shum for help during filming, W. Seah and Pan Ocean Aquarium for assistance in acquiring live specimens, and members of the Wainwright Lab for discussions at early stages of this research.

## Author contributions

**Conceptualization:** Nick Peoples, Michalis Mihalitsis, Peter C. Wainwright.

**Data curation:** Nick Peoples.

**Formal analysis:** Nick Peoples, Michalis Mihalitsis.

**Funding acquisition:** Peter C. Wainwright.

**Investigation:** Nick Peoples, Michalis Mihalitsis.

**Methodology:** Nick Peoples, Michalis Mihalitsis.

**Project administration:** Peter C. Wainwright.

**Resources:** Peter C. Wainwright.

**Supervision:** Peter C. Wainwright.

**Visualization:** Nick Peoples, Michalis Mihalitsis.

**Writing – original draft:** Nick Peoples.

**Writing – review & editing:** Nick Peoples, Michalis Mihalitsis, Peter C. Wainwright.

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
