## [Editor Report · Decision Letter 0]

Dear Dr Peoples,

Thank you for submitting your manuscript entitled "Incompatibility between two major innovations shaped the diversification of fish feeding mechanisms" for consideration as a Research Article by PLOS Biology.

Your manuscript has now been evaluated by the PLOS Biology editorial staff, as well as by an academic editor with relevant expertise, and I'm writing to let you know that we would like to send your submission out for external peer review.

IMPORTANT: We think that it would be better to review your manuscript as a Short Report article. Because your paper is already relatively concise, no re-formatting is needed, but please select "Short Reports" as the article type when you upload your additional metadata (see next paragraph).

Once your full submission is complete, your paper will undergo a series of checks in preparation for peer review. After your manuscript has passed the checks it will be sent out for review. To provide the metadata for your submission, please Login to Editorial Manager (https://www.editorialmanager.com/pbiology) within two working days, i.e. by Feb 19 2025 11:59PM.

Kind regards,

Roli Roberts

Roland Roberts, PhD

Senior Editor

PLOS Biology

rroberts@plos.org

---

## [Decision Letter · Decision Letter 1]

Dear Dr Peoples,

Thank you for your patience while your manuscript "Incompatibility between two major innovations shaped the diversification of fish feeding mechanisms" was peer-reviewed at PLOS Biology. It has now been evaluated by the PLOS Biology editors, an Academic Editor with relevant expertise, and by two independent reviewers.

You'll see that reviewer #1 is positive about the study, but is concerned about your reliance on “largest tooth” measurements, and suggests using molar, incisor and/or average tooth size because of the prevalence of heterodonty among fish. Their other issues are largely presentational. Reviewer #2 is also very positive, but has a significant list of requests, some of which may need additional analyses, and one of which is identical to reviewer #1’s concern about tooth measurement. S/he also criticises your reliance on a single book and self-citation in your treatment of the extensive literature.

In light of the reviews, which you will find at the end of this email, we would like to invite you to revise the work to thoroughly address the reviewers' reports.

Given the extent of revision needed, we cannot make a decision about publication until we have seen the revised manuscript and your response to the reviewers' comments. Your revised manuscript is likely to be sent for further evaluation by all or a subset of the reviewers.

**IMPORTANT - SUBMITTING YOUR REVISION**

*Re-submission Checklist*

*Published Peer Review*

*PLOS Data Policy*

Sincerely,

Roli Roberts

Roland Roberts, PhD

Senior Editor

PLOS Biology

rroberts@plos.org

REVIEWERS' COMMENTS:

Reviewer #1:

Review for Peoples, N. paper

Overall, I liked the paper and the analysis. I think this is a very interesting idea and topic and I'm glad the authors are tackling it. It is very well written, and the statistics are sound. However, I'm a little concerned about the methodology for measurements of tooth size. Comparing the largest tooth in the jaw no matter its location is concerning. I wonder if the authors were to test their theory on the size on the incisor across all species or just the size of the molars if the same relationships would be found. I think this is a worthwhile paper, I'm just concerned there may be a sampling bias based on the methods of measuring the tooth size. Would the authors be able to run an analysis on just incisor or molar size across all species? I'm also wondering about average tooth size since there's a lot of heterodonty in fishes. Would the authors find the same relationship if they looked at average tooth size? You could also look at the range in tooth size as being an indicator. These are some concerns and questions I have about the paper.

Comments:

Lines 65-66: "Fish primarily capture prey by using a combination of forward swimming, termed 'ram', and suction…" I would use this sentence to describe the difference between body ram and jaw ram. Having this definition in the intro would be helpful

Lines 62-87: I feel as if this paragraph jumps around and is not as fluid as it could be. I think starting off with describing the prey capture then moving to teeth then moving back to suction is confusing. I would just talk about prey capture strategies and then talk about teeth. Or maybe start with teeth and then talk about suction and ram

Line 179: Not sure if "shape ratio" is ever defined in the paper.

Line 223: Just a thought but maybe you could talk about the drag teeth may incur on the slender jaws of fish that have protrusible jaws? Not a necessity but thought it might be a cool avenue to think about.

Figure 2a: what does the shaded area and dotted lines represent?

Reviewer #2:

I applaud the authors on a really interesting and compelling read. I can't recall that many papers that look at this sort of "interference" between trait sets. Cool stuff. My comments and critiques are mostly minor and mostly cover times where I think the authors could do a better (meaning more clear) job of presenting what they are doing and why. I leave it to the Editor(s)' discretion about whether the concerns below merit revision, etc.

Lines 62-87: I understand what the authors are doing, but this paragraph comes across a bit stilted ie with short factual statements repeated without a whole lot timing them all together. Perhaps another approach is a paragraph on teeth, then a paragraph on suction/ram feeding (or the reverse), then a synthesis.

Lines 131-136: I like this idea of comparing two extremes, like cichlids and characiforms, but I wonder if it's a little too extreme of an example. Is it worth incorporating a third clade with jaw kinesis, but with variability in kinesis within the clade (wrasses?)? Comparing cichlids to characiforms is like comparing apples to cucumbers. Moreover, while other parts of the text correct for sample size differences, does this section?

Lines 133-134: I believe some characiforms and cithariniforms do have independently derived jaw protrusion, like the halftooths. See Gery 1962.

Lines 154-157: the authors should explain why they took this binning approach in the first place - what question is this method answering and what should readers expect to find. You could do this by starting the paragraph with a question that sets up the rest of the prose.

Lines 166-167: this is presumably an issue of formatting these sorts of papers with Results prior to Methodology - but it's not clear to me at this point how the authors measured "diversity of kinematic strategies" unless by strategies they just mean suction, body ram, and jaw ram.

Lines 182-183: explain to readers what the authors mean when they say: "threshold model for evolution."

Lines 212-213: I do not think that tooth replacement is most frequently intraosseus in teleosts, my recollection is that this is actually quite rare, thinking of papers by Bemis and colleagues. When looking at the references the authors cite for this, both papers examine single species. So, I couldn't confirm, based on citations by the authors, that this is a factual statement. Perhaps the authors, with a focus on cichlids and characiforms, assumed intraosseus replacement is found more often than it is?

Lines 224-226: I guess I missed it at some point - but how are the authors defining "large" teeth vs. "small" teeth?

Lines 224-261: I found this to be an interesting section, but it spends quite a bit of time discussing data that the authors themselves didn't collect. Recommend condensing this to some pertinent examples for a more concise section. Moreover, several times claims are stated about the morphology or ecology of one fish group or another, but aren't referenced. The authors cite Berkovitz & Shellis' book on teeth quite frequently, as a bit of a catch-all reference, without

Lines 268-269: I understand what the authors are saying about "negative implications for resource processing" but again, the authors didn't actually quantify deficits in prey processing, so it seems like a bit of a reach. I imagine that many of these fishes might actually not suffer any deficits, if they have capable pharyngeal jaws. I acknowledge that's dodging a constraint through use of an additional tool, but worth mentioning the caveat.

Figure 1: examining this figure, I wonder if the sampling strategy here included enough species on the jaw ram or suction end of this ternary plot to make a meaningful comparison. Or if those species exist in the first place. If the latter is the case, then that would invoke some other discussions of constraint perhaps?

Figure 4: this is certainly a preference thing, but these cleared and stained jaws with teeth should be edited to that they have similar saturation, hue, contrast, color-balance, etc. The pufferfish image has a large grey blotch on one side. This figure could be refined with little effort and a bit of time in Photoshop.

Lines 347-351: "We quantified the proportions of these three movements (body ram, jaw ram, suction) to closing the total distance between predator and prey" - how, specifically? Meaning, how did the authors differentiate among these three components for measurements? (Note: I found this later, lines 367-370… perhaps this section should be moved up). However, I'm still curious how the authors managed to distinguish these three components independent of each other.

Lines 352-354: how did authors control or account for differences in kinematic sequences if prey were not kept consistent across studies?

Lines 356-359: was any randomization or inter-rater reliability done to see what the effect of measuring the "best" strike was? It seems to me that with this much leeway over what data the authors were analyzing, there can be some inherent bias in selection of which videos were deemed appropriate for analyses.

Lines 377-391: is the largest tooth really the best option here? Rather than the most anterior or average sized tooth? How might the selection of a different metric of tooth size alter the results?

Lines 406-415: the authors mention high phylogenetic signal in the data, but the Dirichlet analyses are reported as not being able to account for this structure in the data. The authors should discuss the limitations of this approach.

I always find it an interesting exercise to gauge how much of the literature in a manuscript is composed of self-citations. My rough estimate puts 30% of the papers cited in this manuscript as being the product of at least one of the authors.

---

## [Editor Report · Decision Letter 2]

Dear Dr Peoples,

Thank you for your patience while we considered your revised manuscript "Incompatibility between two major innovations shaped the diversification of fish feeding mechanisms" for publication as a Short Report at PLOS Biology. This revised version of your manuscript has been evaluated by the PLOS Biology editors and the Academic Editor.

Based on our Academic Editor's assessment of your revision, we are likely to accept this manuscript for publication, provided you satisfactorily address the following data and other policy-related requests.

a) Please address my Data Policy requests below; specifically, we need you to supply the numerical values underlying Figs 1AB, 2AB, 3AB, S1, S2AB, either as a supplementary data file or as a permanent DOI’d deposition. I note that many of the graphs can possibly be directly plotted from the S1_Data.xlsx file that you have supplied; please can you confirm whether this is correct, and if not, supply the additional underlying values. Please also supply the treefile for Fig 1.

b) Please cite the location of the data clearly in all relevant main and supplementary Figure legends, e.g. “The data underlying this Figure can be found in S1 Data” or “The data underlying this Figure can be found in https://zenodo.org/records/XXXXXXXX

c) Please make any custom code available, either as a supplementary file or as part of your data deposition.

We expect to receive your revised manuscript within two weeks.

*Published Peer Review History*

*Press*

Sincerely,

Roli Roberts

Roland Roberts, PhD

Senior Editor

rroberts@plos.org

PLOS Biology

DATA POLICY:

Regardless of the method selected, please ensure that you provide the individual numerical values that underlie the summary data displayed in the following figure panels as they are essential for readers to assess your analysis and to reproduce it: Figs 1AB (includingtreefile), 2AB, 3AB, S1, S2AB. NOTE: the numerical data provided should include all replicates AND the way in which the plotted mean and errors were derived (it should not present only the mean/average values).

CODE POLICY

DATA NOT SHOWN?

---

## [Editor Report · Decision Letter 3]

Dear Nick,

Thank you for the submission of your revised Short Report "Incompatibility between two major innovations shaped the diversification of fish feeding mechanisms" for publication in PLOS Biology. On behalf of my colleagues and the Academic Editor, Anders Hedenström, I'm pleased to say that we can in principle accept your manuscript for publication, provided you address any remaining formatting and reporting issues. These will be detailed in an email you should receive within 2-3 business days from our colleagues in the journal operations team; no action is required from you until then. Please note that we will not be able to formally accept your manuscript and schedule it for publication until you have completed any requested changes.

Sincerely, 

Roli

Senior Editor

PLOS Biology

rroberts@plos.org